# Bioconjugation of Carbohydrates to Gelatin Sponges Promoting 3D Cell Cultures

**DOI:** 10.3390/biomimetics8020193

**Published:** 2023-05-06

**Authors:** Antonietta Pepe, Antonio Laezza, Angela Ostuni, Alessandra Scelsi, Alessandro Laurita, Brigida Bochicchio

**Affiliations:** 1Laboratory of Protein-Inspired Biomaterials, Department of Science, University of Basilicata, Via Ateneo Lucano, 10, 85100 Potenza, Italy; antonio.laezza@unibas.it (A.L.); alexandra.scelsi@virgilio.it (A.S.); 2Cellular Biochemistry Laboratory, Department of Science, University of Basilicata, Via Ateneo Lucano, 10, 85100 Potenza, Italy; angela.ostuni@unibas.it; 3Microscopy Area, Department of Science, University of Basilicata, Via Ateneo Lucano, 10, 85100 Potenza, Italy; alessandro.laurita@unibas.it

**Keywords:** gelatin, sponges, disaccharide, adhesion motifs, bioconjugation, biomaterials, cell culture

## Abstract

Gelatin sponges are widely employed as hemostatic agents, and are gaining increasing interest as 3D scaffolds for tissue engineering. To broaden their possible application in the field of tissue engineering, a straightforward synthetic protocol able to anchor the disaccharides, maltose and lactose, for specific cell interactions was developed. A high conjugation yield was confirmed by ^1^H-NMR and FT-IR spectroscopy, and the morphology of the resulting decorated sponges was characterized by SEM. After the crosslinking reaction, the sponges preserve their porous structure as ascertained by SEM. Finally, HepG2 cells cultured on the decorated gelatin sponges show high viability and significant differences in the cellular morphology as a function of the conjugated disaccharide. More spherical morphologies are observed when cultured on maltose-conjugated gelatin sponges, while a more flattened aspect is discerned when cultured onto lactose-conjugated gelatin sponges. Considering the increasing interest in small-sized carbohydrates as signaling cues on biomaterial surfaces, systematic studies on how small carbohydrates might influence cell adhesion and differentiation processes could take advantage of the described protocol.

## 1. Introduction

Natural polymers are considered an emerging technological platform for the development of innovative materials for biomedical applications. Their inherent biological properties, chemical versatility, biocompatibility, biodegradability, sustainability, and eco-friendliness render them promising alternative materials in this field in comparison to synthetic polymers [1,2,3,4]. Among them, gelatin sponges have gained particular interest and are widely used in medical practice as hemostatic agents as well as scaffolds for different tissue engineering applications [5,6,7]. Another field that could take advantage of the development of gelatin scaffolds for in vivo and ex vivo studies is cancer biology [8]. 3D cell culture systems developed especially for preclinical drug screening could represent an improved model that better replicates the microenvironmental complexity of the tumor than flat monolayer cultures [9,10].

Gelatin is a natural biopolymer obtained from the partial hydrolysis of collagens, the main protein constituting the connective tissues in vertebrate animals, which has been widely used as a biocompatible material due to its generally recognized safe status by the US Food and Drug Administration (FDA) [11]. Gelatin has been shown to form 3D-ordered macroporous structures for applications in oral drug delivery and regenerative tissue engineering [12]. Many reports showed improved results for gelatin-based scaffolds in bone and cartilage tissue engineering [13,14,15,16]. The application of gelatin into electrospun scaffolds for wound healing has also been reported [17,18,19]. The characteristic triple-stranded helix structure of collagen is partially retained in gelatin. However, denaturation at room temperature—as well as partial degradation—weakens the mechanical properties of gelatin and limits its applications. For the reinforcement of gelatin materials, crosslinking and/or compositing with other materials have often been adopted. Crosslinking to obtain water-insoluble gelatin formulations (hydrogel, sponges, electrospun scaffolds) can be easily prepared by physical, chemical, or enzymatic crosslinking methods [20,21,22,23]. To improve the performance of gelatin-based medical devices, several molecules were chemically grafted to gelatin mainly to improve bioactivity by the addition of small molecules—peptides and carbohydrates. Among them, antioxidant molecules, such as caffeic acid [24], drugs such as simvastatin [25], and peptides such as RGD [26] have been conjugated. In addition to the known adhesive peptides, another class of molecules that has gained a lot of attention in recent years, especially in the domain of cell adhesion, is the carbohydrates (mannose, lactose, galactose, etc.), which are common components of lectins present on the cell surface and involved in cell interactions [27]. Mono-, di- and tri-saccharides have been employed as simple carbohydrate cues and conjugated to several biopolymers [28,29]. Glucosamine was successfully embedded in gelatin/HA cryogel as a biological and physical cue to reduce the dedifferentiation of chondrocytes [30]. Many examples of glycosylated chitosan derivatives were synthesized to improve solubility and promote hepatocyte adhesion proliferation with the aim of maintaining liver-specific functions [31]. Lactose-modified chitosans are often cited for their applications in different fields, such as liver, bone, cartilage, and nerve tissue engineering [32,33,34]. Glycosylation of collagen was performed through many different reaction strategies to produce bioactive scaffolds for tissue engineering [35,36,37]. A few reports describe the glycosylation of gelatin [38]. Lactose and glucose were also conjugated to gelatin by a thermal treatment that favors the Maillard reaction to promote crosslinking [39,40].

Because of the large number of glycoproteins present at the cell surface, carbohydrates may be an alternative way to increase cell interaction with a substrate. In addition, the cost and difficulty of grafting sugar molecules are less than those of grafting a synthetic peptide.

In the present work, two disaccharides, lactose and maltose, have been grafted to gelatin to improve cell interaction and adhesion to gelatin sponges designed for tissue engineering. The porous gelatin sponges were fabricated by a freeze-drying process followed by ethylene diamine (ED)-based chemical crosslinking to obtain two water-resistant sponges, GS-Mx and GS-Lx, decorated with maltose and lactose, respectively. For ease of handling, phenotypic stability, and high proliferative capacity, human hepatoblastoma HepG2 cells were used as an in vitro model to test the biomaterials [41,42]. Seeding of HepG2 cells on GS-Mx and GS-Lx sponges showed high cell viability and improved proliferation compared to pristine gelatin sponges. Because of the large number of glycoproteins present at the cell surface, carbohydrates may be an alternative way to drive cell interaction with a substrate. Considering the increasing interest in small-sized carbohydrates as signaling cues on biomaterial surfaces, the presented reaction scheme is of general application and could be exploited for anchoring any reducing carbohydrate. In this way, the described methodology could promote systematic studies on cell-carbohydrate interactions for defining cell adhesion and differentiation processes useful for designing, for example, bone, liver, and cartilage tissue engineering.

## 2. Materials and Methods

### 2.1. Materials

Type B gelatin (225 Bloom) from bovine skin, *N*-(3-Dimethylaminopropyl)-*N*′-ethylcarbodiimide (EDC), sodium cyanoborohydride (NaCNBH_3_), ethylenediamine (ED), sodium sulfate (Na_2_SO_4_), sodium carbonate (NaHCO_3_), and all the organic solvents were purchased from Sigma Aldrich and used without further purification. 4-*O*-β-D-Galactopyranosyl-D-glucose (Lac) and 4-*O*-α-D-Glucopyranosyl-D-glucose (Mal) were purchased from Alfa Aesar (Heysham, UK).

### 2.2. Conjugation of Disaccharides to Gelatin

The disaccharides, Lac or Mal, were conjugated to gelatin by reductive amination following the procedure described in Roy et al. [43]. Briefly, 0.1 M of Lac or Mal was reacted with 1% (*w*/*v*) gelatin (Gel) in a borate buffer solution (BBS, 100 mM, pH 8.5) in the presence of Na_2_SO_4_ (0.5 M) for 3 h at room temperature (RT). Then, two equiv. of NaCNBH_3_ were added at 0 °C. After the dissolution of the reduction agent, the reaction mixture was incubated at 50 °C for 5 days. After cooling to room temperature, the reaction mixture was dialyzed against NaCl 150 mM solution for 2 days, and against H_2_O for further 2 days and then lyophilized obtaining the Lactose-decorated gelatin (GL) and the Maltose-decorated gelatin (GM).

### 2.3. Determination of the Conjugation Degree

The extent of disaccharide conjugation to gelatin was determined by the reduction of free amino group content after using 2,4,6-trinitrobenzene-sulphonic acid solution (TNBS) as an amino-specific assay.

The sample (Gel, GM, or GL) was dissolved in an aqueous solution of NaHCO_3_ (0.1 M pH 8.5) for 30 min, followed by the addition of the TNBS (0.01% *w*/*v*) solution, and was incubated at 37 °C for 2 h. Later, 6 M of HCl was added to hydrolyze the samples at 60 °C for 90 min. The reaction mixture was diluted with deionized (DI) water and absorbance maxima was measured using a Varian Cary 60 UV/Vis spectrophotometer at a wavelength of 346 nm. A gelatin sample was assumed to contain 100% of the available free amine groups and this value was used to calculate the percentage of remaining free amine groups after the conjugation reaction using the following equation (Equation (1)):(1)$$\frac{n\left({NH}_{2}\right)}{g\left(sample\right)}=\frac{\left(A_{346}\times V\right)}{m\times \varepsilon \times l}$$
where C_NH2_ is the concentration of lysine *ε*-amino groups per *g* of gelatin, *A*_346_ is the absorbance at λ = 346 nm, *V* is the volume (L), *m* is the mass of the *sample* (*g*), *ε* = 14,600 L mol^−1^ cm^−1^ is the molar extinction constant of TNBS–Lys, and *l* is the cell path length (cm). The degree of conjugation was evaluated as the difference between the chemically determined number of free amine groups before and after crosslinking, relative to the initial free amine content.

### 2.4. NMR Spectroscopy

^1^H NMR spectra were recorded on a Varian Unity INOVA 400 MHz spectrometer equipped with a multinuclear probe and z-axial gradients. The samples of Gel, GM, and GL were prepared by dissolving 5 mg in 700 μL of D_2_O containing 0.1 mM of 3-(trimethyl-sily1)-1-propane sulfonic acid (DSS) as internal reference standard at 0 ppm. One-dimensional spectra were acquired in Fourier mode with quadrature detection.

### 2.5. Fourier Transform Infrared (FT-IR) Spectroscopy

The samples for FT-IR spectroscopy were analyzed in the solid state as potassium bromide (KBr) pellets. The lyophilized samples were mixed with KBr to a final concentration of approximately 1 wt%. FT-IR spectra were recorded on a Jasco FTIR-460 spectrometer. Each spectrum is the result of signal-averaging of 256 scans at a resolution of 2 cm^−1^. All spectra are presented as absorbance spectra after background subtraction.

### 2.6. Preparation of the Gelatin Sponges

An aliquot (2 mL) of 1% (*w*/*v*) solutions of Gel, GM, or GL was poured into a 12-well culture plate, frozen overnight at −80 °C and then lyophilized for 5 days to obtain GS, GS-M, and GS-L porous sponges, respectively. Since the resulting sponges are water soluble, they were crosslinked by soaking into 10 mL of acetone/H_2_O (9:1, *v*/*v*) containing 25 mg of EDC for 1 h at room temperature. After activation of the carboxyl group by EDC 670 μL of ED (d = 0.89 g/mL) was added to the reaction mixtures were shaken at 60 rpm using an orbital shaker (PSU-10i, Biosan, Riga, Latvia) for 24 h. The sponges were gently dried by using filter paper and washed with ultrapure water (20 mL × 3 times) to remove any residual trace of crosslinker. The molar ratio of COOH: EDC:ED was calculated to be 1:6.5:0.5. The resulting crosslinked sponges, GS-x, GS-Mx, and GS-Lx, were thoroughly washed with MilliQ water and re-lyophilized.

### 2.7. Swelling Test

GS-x, GS-Mx, and GS-Lx sponges were weighed in air-dry conditions. Then they were soaked in deionized water for 1 h. Wet samples were wiped with filter paper to remove excess liquid and re-weighed. The swelling ratio (*SW*) was calculated as (Equation (2)):(2)$$SW=\frac{W_{w}-W_{d}}{W_{d}}$$
where *W_w_* and *W_d_* are the weights of the wet and the dry samples, respectively. The experiments were repeated five times.

### 2.8. Scanning Electron Microscopy (SEM)

The internal structure of the sponge samples before and after crosslinking was observed using a scanning electron microscope (Philips-Fei ESEM XL30-LaB6). Prior to imaging, the freeze-dried sponges were freeze-fractured after immersion in liquid nitrogen and then mounted using carbon tape on aluminum SEM stubs and sputtered with a thin gold layer.

### 2.9. Cell Culture

Human hepatoblastoma cells (HepG2) were grown in high glucose (4.5 g/L) Dulbecco’s modified Eagle’s medium (DMEM), supplemented with 10% fetal bovine serum (FBS), 2 mM L-glutamine, 100 U/mL penicillin, and 100 μg/mL streptomycin. Cell cultures were maintained at 37 °C in a water-saturated atmosphere with 5% CO_2_.

### 2.10. Cell Counting

HepG2 cells (0.2 × 10^6^) were seeded onto a 12-well plate on the surface of previously UV-light sterilized sponges and cultured at 37 °C in the incubator. Cell viability was monitored at 24 h, 48 h, and 72 h by the trypan blue exclusion assay [44]. As a control, HepG2 cells were seeded at the same density directly on polystyrene plates.

### 2.11. Cell Adhesion

In a 12-well plate, at a density of 0.1 × 10^6^ cells/300 μL, cells were seeded on sponges previously pre-wetted in DMEM and UV-sterilized overnight in a laminar flow hood. After 48 h, the sponges were transferred to a new 12-well plate with 1 mL of fresh medium, which was replaced with fresh medium every 3 days. After 13 days, the sponges were rinsed with PBS and fixed with 1 mL of 10% formaldehyde solution before 0.1% crystal violet staining for 15 min. After staining, we observed the images through a phase-contrast microscope (Nikon Eclipse, TS100, Tokyo, Japan).

### 2.12. SEM

The morphology of the HepG2 cells was examined by SEM. After fixation in 2.5% (*v*/*v*) glutaraldehyde in pH 7.5 phosphate buffer, cell-seeded sponges were dehydrated in a graded ethanol solution series (from 20% to 100%). Samples were then critical-point dried using liquid CO_2_ and sputtered with a thin gold layer before examination under a Philips-Fei ESEM XL30-LaB6 scanning electron microscope.

### 2.13. Statistical Analysis

Data are presented as mean ± SD. Statistical analyses were performed using OriginPro 2015 (9.2) software. Statistical significance was evaluated using one-way ANOVA. Post hoc multiple comparisons were determined by the Tukey test with the level of significance set at * *p* < 0.05.

## 3. Results and Discussion

### 3.1. Conjugation of Disaccharides Maltose and Lactose to Gelatin

In the present study, a straightforward protocol was developed that permits (i) the anchoring of disaccharides to gelatin by reductive amination, employing the lysine ε-amino groups (Figure 1), and (ii) the crosslinking of the decorated gelatin by amide formation, exploiting the carboxylic acids functionality of the side chain of Glu and Asp amino acids present in the gelatin sequence (Figure 2).

Two kinds of modified gelatin derivatives, namely, GM and GL, were prepared by grafting maltose and lactose, respectively. The conjugation of carbohydrates to gelatin was performed by a chemical protocol developed by Roy et al. with slight modifications [43]. Gelatin 1% (*w*/*v*) was added to a solution of 100 mM of disaccharide dissolved in borate buffer (pH 8.5) at 37 °C to avoid gelation. According to studies by Gildersleeve et al., borate buffer was the most effective in improving conjugation together with the addition of 0.5 M Na_2_SO_4_ [45]. Finally, after 3 h, a reduction agent, NaCNBH_3_, was added and the reaction was heated to 50 °C for 5 days. Lactose and maltose are both reducing disaccharides and were able to form iminium ions through their free reducing aldehyde group reacting with lysine ε-amino groups. Gelatin has a significant number of lysine residues (3.0 × 10^−4^ mol/g of gelatin) allowing it to conjugate a considerable number of molecules. The extent of functionalization was determined by measuring the amount of free or unreacted amino groups in the glycosylated gelatin samples by TNBS test. A high conjugation degree was observed in the proposed reaction conditions for both GL (95 ± 5%) and GM (90 ± 6%).

### 3.2. Characterization of GM and GL

The gelatin derivatives were analyzed by ^1^H-NMR spectroscopy acquiring proton spectra of the dialyzed GM and GL in D_2_O. The ^1^H-NMR spectrum of GM (Figure 1, blue curve) confirmed the conjugation with gelatin by the presence of broad signals related to the protein backbone (Figure 1, red curve) and to signals at δ_DSS_ = 5.12 ppm and δ_DSS_ = 3.43 ppm, respectively associated to the C*H*-1 and C*H*-2 of the open form of aminated D-glucose, both upfield shifted respect to the starting material (Figure 1, black curve) [46].

In the case of GL, the ^1^H-NMR spectrum (Appendix A, blue curve) confirmed the conjugation with gelatin by the presence of wide signals related to the protein backbone, as well as to the presence of wide peaks at δ_DSS_ = 3.76 ppm and δ_DSS_ = 3.55 ppm that fall in the same region of lactose carbinolic protons (Appendix A, black curve) and are absent in the pure gelatin (Appendix A, red curve).

Further characterization was carried out by FT-IR spectroscopy, which allowed the assignment of specific functional groups in the spectra of gelatin, maltose, lactose, and the conjugated-gelatin derivatives GM and GL (Figure 2a,b). The FT-IR spectrum of Gel (red curve) showed typical bands of proteins: the amide A band, at about 3521 cm^−1^, assigned to N–H stretching; at approximately 3316 cm^−1^, a broad band that originated from O–H stretching of 4-hydroxyproline residues. At shorter wavenumbers, the amide I band at 1652 cm^−1^ ascribed to gelatin C=O stretching and the amide II band situated at 1551 cm^−1^ due to N–H bending were visible. Lactose and maltose spectra, in Figure 2a and Figure 2b, respectively, (black curve) are dominated by a broad band in the range of 3700 to 3100 cm^−1^ assigned to the stretching vibrations of carbohydrates’ O–H groups. Furthermore, bands due to aliphatic C–H stretching at about 2900 cm^−1^ and to C–O stretching, typical of carbohydrates, were observed in the region ranging from 1200 to 1100 cm^−1^. The FT-IR spectra of the GM and GL derivatives (blue curve) show the typical bands observed for gelatin together with bands ascribed to the disaccharides lactose and maltose. In order to aid comparison, the spectra of Gel and GM, and Gel and GL, were height-normalized based on the amide I band at 1652 cm^−1^. In particular, in the region 1200–1100 cm^−1^ where bands corresponding to C–O stretching are located, more intense signals are observed for the conjugated gelatins. Increased intensity is observed also for the O–H and N–H bands in the 3500–3300 cm^−1^ region. These spectral features are ascribed to the successful conjugation of the disaccharides to gelatin in GM and GL.

### 3.3. Preparation and Characterization of Porous Sponges from GM and GL

Gelatin and its derivatives, GM and GL, are highly soluble in water and were processed to form porous sponges by freeze-drying obtaining white porous sponges GS, GS-M, and GS-L, respectively (Appendix A). Gelatin scaffolds exhibit weak mechanical strength and poor hydrolysis resistance. To overcome this issue, gelatin scaffolds are usually stabilized by crosslinking to increase their strength and hydrolysis resistance and maintain their stability during implantation [47]. To avoid the collapse of the porous structure when the sponges are soaked in aqueous solutions, a crosslinking reaction was carried out by EDC activation of the carboxylic groups of gelatin in acetone/H_2_O (9/1) (Figure 2). The acetone/H_2_O (9/1) solution was chosen because it preserved the porous structure of the sponges while solubilizing reactants. Ethylenediamine (ED) was employed as the crosslinking agent, reacting with the carboxylic groups present in the side chains of Glu and Asp of gelatin.

The morphology of the resulting sponges was investigated by SEM microscopy. In Figure 3, SEM images of GS, GS-L, and GS-M sponges acquired before crosslinking showed a highly porous structure on the surface and inside the sponge as revealed after freeze-fracturing. GS-M and GS-L showed some differences in the surface’ structural features with a more dense and continuous organization with limited fenestrations. The inner structures of all the samples were constituted by highly homogeneous and interconnected pores without any appreciable differences between the sponges.

After crosslinking (Figure 4), the GS-x, GS-Mx, and GS-Lx sponges showed again a great variation on the surface of the sponges, with partial closing of the porous structure probably by coalescing, while in the internal structure of the sponge, limited change in the morphology was observed. EDC crosslinking conditions preserved the porous structure avoiding the collapse of void structures and was preferred to glutaraldehyde vapor treatment that caused fusing of the pores [47].

Swelling of the GS-x, GS-Mx, and GS-Lx sponges was tested by soaking the sponges in water for 1 h and weighing the uptaken water. The average swelling of the scaffolds, expressed in terms of swelling ratio, was 37.2 ± 2.0, 24.7 ± 1.5, and 24.4 ± 7.4 for the GS-x, GS-Mx, and GS-Lx sponges, respectively (Figure 5). The capacity to absorb fluids from the surrounding cell environment is an important factor because it can define the retention of physiological fluids in vivo, which results in improved cell infiltration and attachment into the scaffolds [48]. Surprisingly the swelling ratios of disaccharide-conjugated sponges are significantly lower than those of the pristine gelatin sponges, even if carbohydrates usually increase hydrophilicity and consequently wettability and water uptake in the scaffolds [49]. Differences in porosity observed on the surface of the sponges could be responsible for the variation in the swelling properties of the sponges [48]. Comparing data of the sponges with cryogels obtained by EDC or GA crosslinking during freezing at −16 °C and, finally, a thawing step, we observed a higher swelling ratio of the scaffolds obtained by the proposed protocol [50,51].

### 3.4. HepG2 Cells Cultured on GS, GS-L, and GS-M Sponges

To investigate the viability of HepG2 cells on different scaffolds, live/dead cells were monitored at 24 h, 48 h, and 72 h by the trypan blue exclusion assay. None of the scaffolds impaired cell viability Furthermore, no morphological variation was observed. To verify cell adhesion, HepG2 cells were left to grow on sponges for up to 17 days. The cells had anchored to the supports and grown. Crystal violet staining showed a higher number of cells on GS-Lx and GS-Mx sponges compared to those on GS-x (Appendix A).

### 3.5. Cell Morphology on the Scaffolds

The detailed cellular structure of HepG2 cells grown on different scaffolds was observed by scanning electron microscopy after 13 days of seeding. As already observed on sponges stained with crystal violet, SEM analysis showed few cells on the gelatin GS-x sponges (Figure 6a,b) allowing us to hypothesize that the cells, having poor adhesion to the scaffold, are unable to form a cluster essential for their growth. On the GS-Mx and GS-Lx scaffolds, cells were distributed over the entire surface, developing tight cell-cell interactions. HepG2 cells were smaller in size on the GS-Mx sponges, retaining a spherical morphology and tending to form clusters (Figure 6c,d) while cells exhibited a more flattened shape on GS-Lx sponges (Figure 6e,f). Cells grown on both decorated sponges showed microvilli-like structures in GS-Lx and GS-Mx sponges (Figure 6g,h respectively). The presence of galactose in GS-Lx sponges could be responsible for the observed morphology and higher density of cytoskeletal filaments, facilitating the asialoglycoprotein-mediated cell adhesion of HepG2 cells [52,53,54]. Spreading of cells on GS-Lx demonstrated an improved cell–matrix interaction, while on GS-Mx sponges the spherical morphology of the HepG2 cells suggested a different anchoring to the scaffold, probably due to the presence of the type-1 glucose transporter (GLUT-1) receptor [55]. GLUT-1 is highly expressed also in chondrocytes [56] and could be involved in cell-scaffold interactions in glucose-carrying scaffolds. The presence of D-glucose on the GS-Mx sponge could provide a suitable cell microenvironment able to maintain the chondrocytic phenotype [57]. Further investigations on this aspect are in progress.

## 4. Conclusions

One of the major advantages of sponge scaffolds is their high porosity, contributing to cell growth and promoting the exchange of nutrients and metabolites. Furthermore, fundamental requirements for scaffolds designed for tissue engineering are the absence of cytotoxicity and the promotion of adhesion of cells in order to produce a new matrix for the developing tissue. In this framework, we proposed a simple and effective carbohydrate decoration protocol to promote bioactive gelatin-based sponges with high viability and improved adhesion of HepG2 cells. The synthetic strategy of lactose and maltose conjugation took advantage of the aldehyde moiety of disaccharides that has been conjugated in an efficient way to gelatin, by a reductive amination reaction. Lactose decoration of scaffolds appending bioactive galactose units is widely exploited for liver tissue engineering, as well as for cartilage regeneration. Less well-characterized are other types of carbohydrate decoration, such as maltose and mannobiose, appending glucose and mannose units, respectively, on the gelatin scaffolds on the adhesion, proliferation and differentiation of different cell types. Studies on this aspect could take advantage of the straightforward protocol described in the present work.

## Data Availability

Data are contained within the article or Appendix A.

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
