# Peer review of "Bioconjugation of Carbohydrates to Gelatin Sponges Promoting 3D Cell Cultures"

_biomimetics, 2023, doi:10.3390/biomimetics8020193_

Round 1
Reviewer 1 Report
1.
The authors should introduce the research of 4D cell culture using gelatin material. In the current version, the readers cannot understand the novelty. Therefore, in the introduction or discussion section, the authors should describe the novelty of this study by comparing it with related papers.
Overall (review not limited to gelatin)
Tissue Eng. Part B Rev. 2010, 16, 351–359.
Cancers 2020, 12(10), 2754
Research papers
Polym. Chem., 2019, 10, 3180-3193
J. Mater. Chem. B, 2019, 7, 1064-1075
http://doi.org/10.1089/ten.tec.2019.0189
2.
The degradation profile of sponges should be investigated. The profile must affect the function of 3D culture.
3.
How about the stiffness? This property also would affect the function.
Good.
Reviewer 2 Report
The introduction and discussion part need to improve.

Reviewer 3 Report
The paper is very well organized, the authors demonstrating a good approach, presenting the topic from several points of view and good concordance between the sections. However, the Introduction does not provide sufficient background, so literature is still needed.
When you present the conjugation of disaccharides to gelatin process, on line 93, specify that room temperature will be abbreviated RT in order to understand the description in Scheme 1 and Scheme 2.
Regarding “2.6. Preparation of the gelatin sponges:”, the authors described in detail the cross-linking process, but how exactly did you choose "10 ml of acetone/H2O (9:1, v/v) containing 25 mg of EDC......670 ml of ED (d=0.89 g/ml)…”?
Pay attention to the writing of the words: crosslinking or cross-linking; crosslinked or cross-linked. Keep them in the same form throughout the manuscript.
About units, use the international system of units (SI). For example, in “2.6. Preparation of the gelatin sponges:” and “2.7. Swelling test:”, the unit for time was not written correctly. Please, see Lines 134, 135, 140.
In “3.2. Characterization of GM and GL”, all FT-IR spectra are presented as absorbance spectra, but it should be specified if there are arbitrary absorbance units.
The authors need to eliminate grammatical and spelling errors.
This manuscript could be accepted after a minor revision.
The authors need to eliminate grammatical and spelling errors.
Reviewer 4 Report
The work is well conceived, however I suggest to:
1. improve the state of the art in the introduction section
2. Figure 3 and 4 resolution must be improved
The work is well written, minor English editing are required.
Round 2
Reviewer 2 Report
Accept the Manuscript.